# ACF7 regulates inflammatory colitis and intestinal wound response by orchestrating tight junction dynamics

Yanlei Ma[1,2,3,*], Jiping Yue[2,*], Yao Zhang[4,*], Chenzhang Shi[1,*], Matt Odenwald[5], Wenguang G. Liang[2], Qing Wei[6], Ajay Goel[7], Xuewen Gou[2], Jamie Zhang[2], Shao-Yu Chen[8], Wei-Jen Tang[2], Jerrold R. Turner[9], Feng Yang[3], Hong Liang[3], Huanlong Qin[1] & Xiaoyang Wu[2]

In the intestinal epithelium, the aberrant regulation of cell/cell junctions leads to intestinal barrier defects, which may promote the onset and enhance the severity of inflammatory bowel disease (IBD). However, it remains unclear how the coordinated behaviour of cytoskeletal network may contribute to cell junctional dynamics. In this report, we identified ACF7, a crosslinker of microtubules and F-actin, as an essential player in this process. Loss of *ACF7* leads to aberrant microtubule organization, tight junction stabilization and impaired wound closure *in vitro*. With the mouse genetics approach, we show that ablation of *ACF7* inhibits intestinal wound healing and greatly increases susceptibility to experimental colitis in mice. *ACF7* level is also correlated with development and progression of ulcerative colitis (UC) in human patients. Together, our results reveal an important molecular mechanism whereby coordinated cytoskeletal dynamics contributes to cell adhesion regulation during intestinal wound repair and the development of IBD.

[1] Department of GI surgery, Shanghai Tenth People's Hospital Affiliated with Tongji University, 301 Yanchang Road, Shanghai 200072, China. [2] The University of Chicago, Ben May Department for Cancer Research, Chicago, Illinois 60637, USA. [3] State Key Laboratory Cultivation Base for the Chemistry and Molecular Engineering of Medicinal Resources, Ministry of Science and Technology of China, Guanxi Normal University, Guilin 541004, China. [4] Department of Colorectal Surgery, Fudan University Shanghai Cancer Center, Shanghai, China. [5] Pritzker School of Medicine, University of Chicago, Chicago, Illinois 60637, USA. [6] Department of Pathology, Shanghai Tenth People's Hospital Affiliated with Tongji University, 301 Yanchang Road, Shanghai 200072, China. [7] Center for Gastrointestinal Research, Center for Epigenetics, Cancer Prevention and Cancer Genomics, Baylor Scott & White Research Institute and Charles A. Sammons Cancer Center, Texas, USA. [8] Department of Pharmacology and Toxicology, University of Louisville Health Science Center, Louisville, Kentucky 40292, USA. [9] Departments of Pathology and Medicine (GI), Brigham and Women's Hospital, Harvard Medical School, Boston, Massachusetts, USA. * These authors contributed equally to this work. Correspondence and requests for materials should be addressed to H.L. (email: hliang@mailbox.gxnu.edu.cn) or to F.Y. (email: fyang@mailbox.gxnu.edu.cn) or to H.Q. (email: hl-qin@hotmail.com) or to X.W. (email: xiaoyangwu@uchicago.edu).

Cell/cell junctions are dynamic processes that are tightly regulated in response to cellular context and signalling[1]. Remodelling of cell/cell junctions entails the interplay between a range of fundamental cellular processes, including cytoskeletal reorganization, surface presentation and internalization of receptors and intracellular trafficking[2]. Among different cell adhesions, tight junctions anchor to the F-actin, and previous studies have also demonstrated its connection with the microtubule network[3]. Microtubules and F-actin control a variety of cellular functions, and accumulating evidence suggests that the coordinated dynamics of microtubule and F-actin is particularly important for cell migration and adhesion, which are intrinsic processes and essential features of tissue morphogenesis, physiology and homeostasis[4,5]. Previous studies have suggested that disruption of microtubules can stabilize tight junction *in vitro* by delaying its disassembly induced by depletion of extracellular calcium[6]. However, it remains unclear how cytoskeletal dynamics is orchestrated at tight junctions and how the coordination may control their dynamics. In this study, we have made major inroads into understanding this process by identifying ACF7 (actin crosslinking factor 7)/MACF1 (microtubule and actin crosslinking factor 1)/macrophin as a key player in this process.

ACF7 belongs to the evolutionarily conserved spectraplakin family of proteins[7]. Although expression of spectraplakins is broad in different tissues, spectraplakins' functions are best characterized in muscle, neurons and epithelial tissues. Mutations in the single *Drosophila* spectraplakin leads to a wide variety of defects, including aberrant cytoskeletal organization, cell/cell junction and integrin-mediated epidermal attachment[7]. There are two spectraplakins in the mammalian genome, BPAG1 (Bullous Pemphigoid Antigen 1) and ACF7. BPAG1 mutant mice display sensory neuron and muscle degeneration phenotype. Loss of BPAG1 also causes gross defects in cytoskeletal organization and function[8,9]. By contrast, deletion of ACF7 expression in mice results in early embryonic lethality[10,11]. cKO of ACF7 suggests that ACF7 is critically involved in cytoskeletal dynamics, cell adhesion and cell migration *in vivo* in different tissue and organs[12–17]. With skin as a model system, we have demonstrated that microtubule and F-actin coordination mediated by ACF7 can guide microtubule plus ends towards focal adhesions, the cellular organelle that mediates cell adhesion to extracellular matrix[16–18]. ACF7-guided targeting of microtubules can enhance turnover of focal adhesions and thus promote cell motility, a process essential for epithelial wound repair in skin epidermis[16,17].

The mammalian intestinal epithelium is a dynamic system in which various biological processes, including intestinal barrier function and cell motility, are coordinated to maintain tissue homeostasis and injury response[19–22]. Accumulating evidence indicates that cell/cell junctions, particularly tight junctions, are critically involved in epithelial barrier function, and the aberrant regulation of tight junctions contributes to pathogenesis in the digestive system, such as inflammatory bowel disease (IBD) (refs 1,23–26). However, the molecular mechanisms underlying the regulation of tight junctions during intestinal tissue homeostasis and the pathogenesis of IBDs remain elusive.

In this report, we demonstrate that depletion of endogenous ACF7 in intestinal epithelial cells leads to deregulation of cytoskeletal and tight junction dynamics and impairs the healing of intestinal wound *in vitro*. To address the molecular mechanism, we show that ACF7 is essential for orchestrating microtubule dynamics in intestinal epithelial cells. By X-ray crystallography, we further delineate the structural basis underlying ACF7 interaction with microtubules, and corroborate the role of this interaction in tight junction dynamics and wound healing. With conditional gene targeting to ablate ACF7 expression in the intestine of mice, our findings have uncovered essential roles for ACF7 in intestinal tissue morphogenesis and epithelial cell migration *in vivo*. In addition, the conditional knockout (cKO) animals are susceptible to experimental colitis, and the expression of ACF7 is strongly correlated with development of ulcerative colitis (UC) in human patients. Taken together, with a comprehensive approach ranging from mouse genetics to molecular and cell biology, our work identified a hitherto unappreciated mechanism by which tight junction dynamics and intestinal wound repair are regulated by cytoskeletal coordination.

## Results

**ACF7 regulates tight junction dynamics and wound healing.** ACF7 is a key crosslinker of microtubule and F-actin networks. To probe its potential role in the intestinal epithelium, we first developed an *in vitro* model of ACF7 knockout (KO) with CRISPR (clustered regularly-interspaced short palindromic repeat) technology[27]. A colorectal adenocarcinoma cell line (Caco-2) was infected with lentivirus encoding Cas9 (CRISPR associated protein 9) and an ACF7-specific gRNA (guide RNA). Clones of infected cells were selected upon antibiotics treatment, and ablation of ACF7 was confirmed by the western blotting analysis (Fig. 1a). Deletion of ACF7 has been shown to inhibit cell motility in various systems *in vitro* and *in vivo*[12–17]. With ACF7-null Caco-2 cells, we analysed cell movement capability with both a scratch-wound model and an oligocellular wound model induced by laser ablation *in vitro*. Our results demonstrated significantly impaired cell motility *in vitro* upon loss of ACF7 (Fig. 1b,c).

Cell adhesion dynamics is critically involved in cell motility[28]. In cultured skin epidermal progenitor cells, ACF7 exhibits enriched localization at focal adhesions, where ACF7 mediates specific crosslinking of microtubule plus ends with F-actin filaments and promotes dynamics of focal adhesions[16]. In addition, a recent study demonstrated that deletion of ACF7 can affect colonic permeability and reduce the expression of tight junction proteins in intestinal epithelium *in vivo*[29]. Interestingly, staining of endogenous ACF7 in Caco-2 cells shows significant co-localization of ACF7 with cell/cell junctions, particularly tight junctions (Fig. 1d). Like focal adhesions, tight junctions are associated with both F-actin filaments and microtubules[3,30]. It has also been shown that the disassembly of microtubule network in various epithelial cell lines leads to stabilization of tight junctions[6], resembling the behaviour of focal adhesions in response to microtubule disruption[16,31,32]. To test the hypothesis that ACF7-mediated microtubule and F-actin crosslinking is also involved in the dynamic remodelling of tight junctions, we induced disassembly of tight junctions in Caco-2 cells by depletion of extracellular calcium. In wild-type (WT) cells, depletion of calcium led to a rapid disruption of tight junctions within 1 h. By contrast, loss of ACF7 or disruption of microtubule network by treatment with nocodazole significantly inhibited the disassembly of tight junctions induced by calcium removal (Fig. 1e).

To further characterize tight junction dynamics in ACF7-deficient cells, the cells were transfected with expression plasmids encoding mRFP-ZO1. WT Caco-2 cells displayed robust fluorescence recovery in the FRAP (Fluorescence Recovery After Photobleaching) test, and treatment of cells with nocodazole or KO of ACF7 dramatically delayed the recovery (Fig. 1f). Quantifications demonstrated significantly elongated $T_{1/2}$ (half time) for fluorescence recovery in both nocodazole-treated cells and ACF7 KO cells (Fig. 1g), suggesting a critical role of an intact microtubule network and ACF7-mediated microtubule crosslinking in tight junction dynamics.

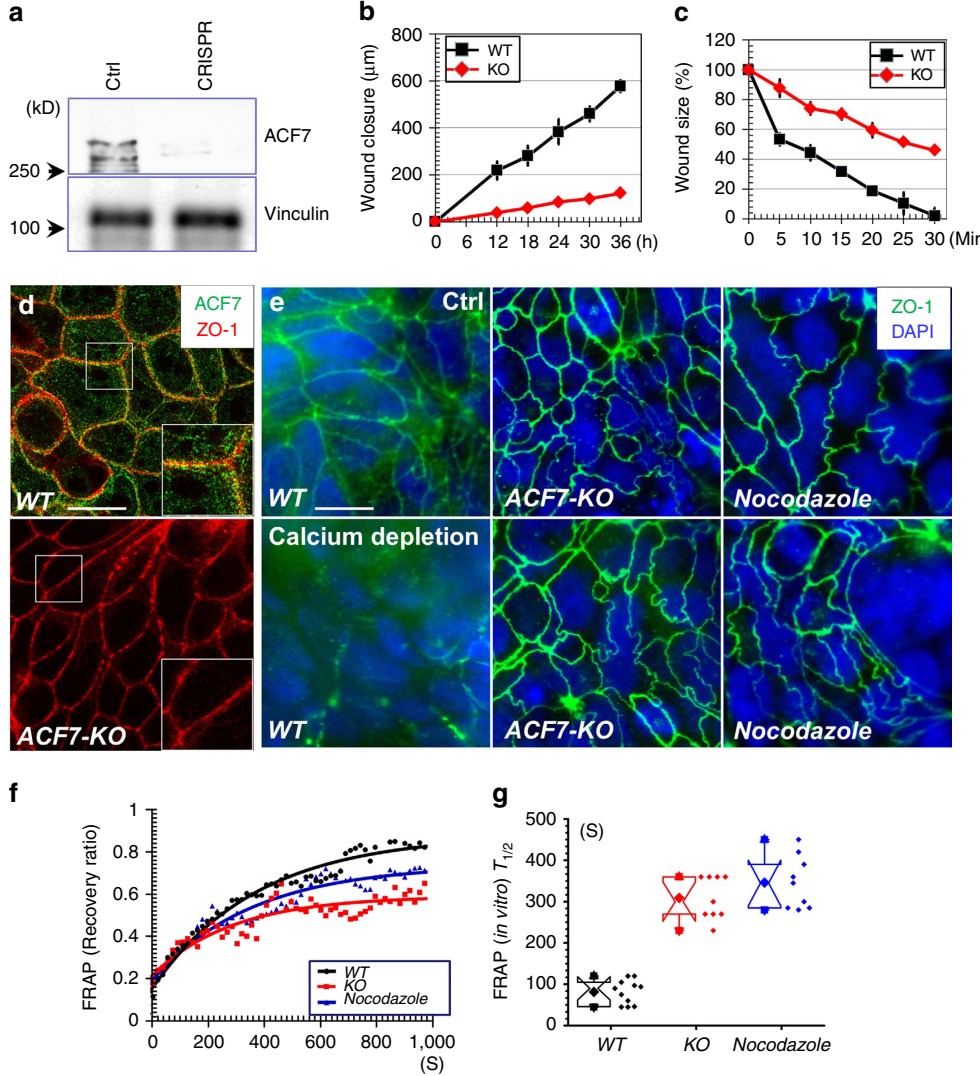

**Figure 1 | ACF7 regulates motility and tight junction dynamics in colorectal epithelial cells.** (**a**) Western blot analysis verified CRISPR-mediated deletion of *ACF7* in Caco-2 cells. Blot of vinculin servers as a loading control. (**b**) Confluent monolayer of Caco-2 cells was subjected to *in vitro* scratch-wound assay. Quantification of the kinetics of wound closure shows slower movement of *ACF7*-deficient cells. Error bar represents standard deviation (s.d.). Sample size $n = 3$ (three independent tests). (**c**) Oligocellular wounds were introduced to monolayer of WT or *ACF7*-null Caco-2 cells. Quantification of the kinetics of the wound closure shows impaired wound healing upon loss of *ACF7.* Error bar represents s.d. Sample size $n = 3$ (three independent tests). (**d**) WT or *ACF7* KO caco-2 cells were subjected to immunofluorescence staining with different antibodies as indicated. Boxed areas are magnified as insets. Scale bar, 50 μm. (**e**) WT or *ACF7* KO cells or WT cells treated with nocodazole were stained for tight junctions with antibody against ZO-1, before or after calcium depletion. Cell nuclei are counter-stained with DAPI. Scale bar, 50 μm. (**f**) *In vitro* FRAP analysis of mRFP-*ZO1* in WT cells, nocodazole-treated WT cells and *ACF7*-deficient Caco-2 cells. Experiments repeated for more than three times. (**g**) Box and whisker plotting of $T_{1/2}$ for FRAP analysis *in vitro*. The box and whisker plot indicates the mean (solid diamond within the box), 25th percentile (bottom line of the box), median (middle line of the box), 75th percentile (top line of the box), 5th and 95th percentile (whiskers), first and 99th percentile (solid triangles) and minimum and maximum measurements (solid squares). Deletion of *ACF7* and nocodazole treatment significantly increase $T_{1/2}$ ($P < 0.01$, Student's *t*-test). Sample size $n > 9$ (three independent tests, and three technical replicates each).

**ACF7 orchestrates microtubule dynamics in intestinal cells**. ACF7 is a key molecule that crosslinks microtubule and F-actin networks in mammalian cells[22,33,34]. Consistent with a previous report[3], tight junctions in WT Caco-2 cells are closely associated with the dense microtubule network, as determined by confocal imaging (Fig. 2a). Ablation of *ACF7* did not cause an overall decrease in the polymerized microtubule network; however, confocal imaging often revealed significant gaps between the microtubule network and the tight junctions (Fig. 2a, and Supplementary Movies 1 and 2). The separation of the microtubule network and tight junction can also be visualized in the 2D surface plot of the ZO-1-positive tight junction ring structure with the underlying microtubules (Fig. 2b). Whereas ZO-1 and α-tubulin staining exhibited a significant overlap in the 2D plot in WT cells, deletion of *ACF7* significantly increased the gap between ZO-1 and the microtubule layer (Fig. 2b and quantification in Fig. 2c). Together, these results strongly suggest that ACF7 plays an important role in mediating microtubule interaction at tight junctions in intestinal epithelial cells.

The carboxy terminus of ACF7 contains the GAR domain, which is responsible for microtubule binding[22,33,34]. However, it remains mechanistically unclear how ACF7 associates with microtubules and how this interaction is regulated in cells. To address this issue, we isolated the ACF7-GAR protein and

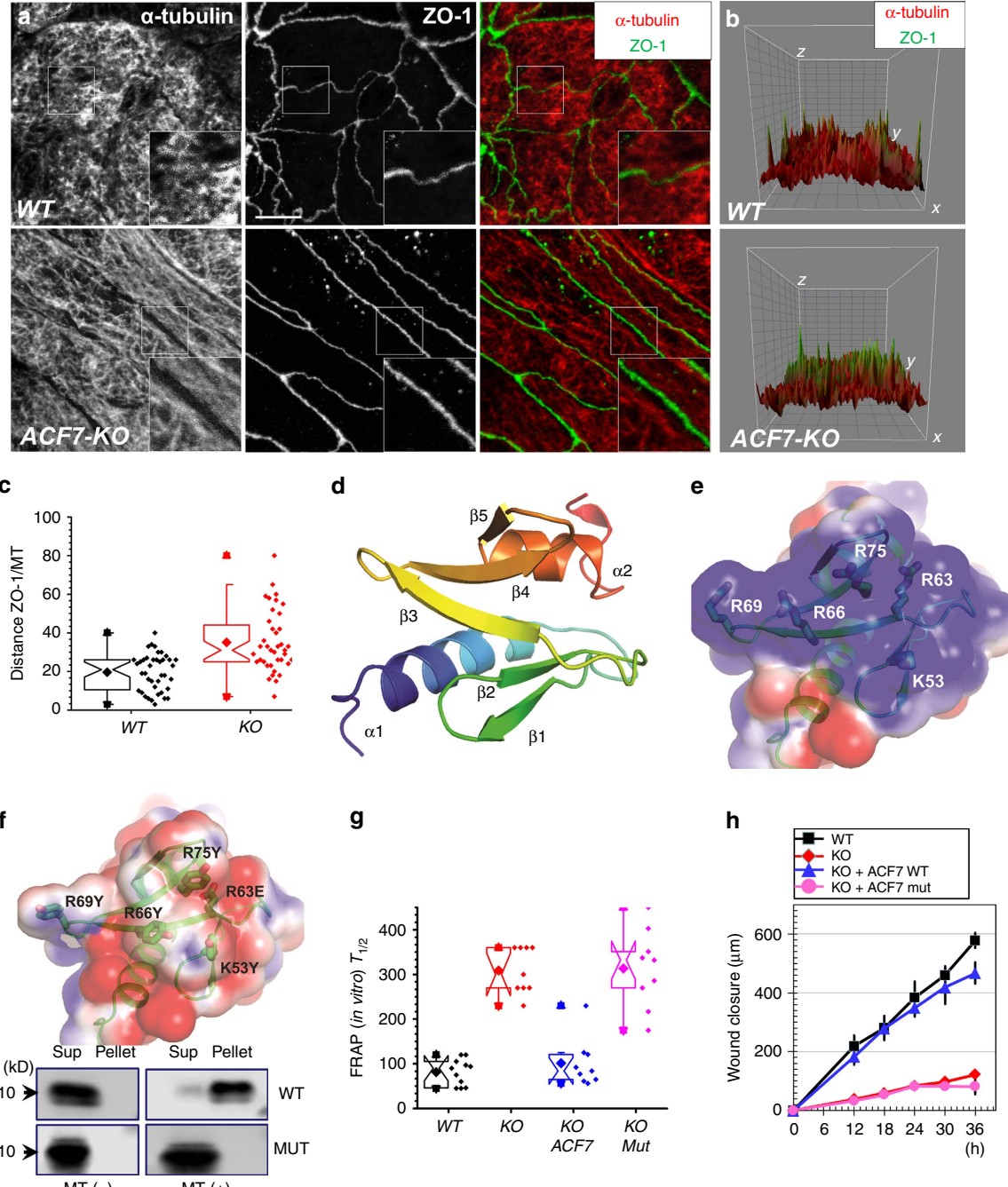

**Figure 2 | ACF7 regulates microtubule network in colorectal epithelial cells.** (**a**) WT and *ACF7* KO cells were subjected to immunofluorescence staining with different antibodies as indicated. Scale bar, 20 µm. Boxed areas are magnified as insets. (**b**) Potential co-localization of ZO-1 and microtubules was determined by surface plot in the representative tight junctions. The surface plot is presented in 3D with $X/Y$ plane at the bottom. (**c**) Quantification of the gap between tight junction (ZO-1) with microtubule network in WT and *ACF7* KO cells. Note significantly increased gap upon loss of *ACF7* ($P < 0.05$, Student's $t$-test). Sample size $n > 30$ (three independent tests, and more than 10 technical replicates each). (**d**) Ribbon representation of ACF7-GAR domain structure. The structure is coloured in rainbow from N-terminus to C-terminus. (**e**) Electrostatic surface potential of WT GAR domain. The electrostatic potential from red ($-1kT$) to blue ($+1kT$) is coloured as calculated by APBS (ref. 67) after prepared by PDB2PQR server[68]. Note WT GAR has a positively charged surface, with contributions from five positively charged residues. (**f**) Mutation of the five positively charged residues on GAR surface alters the electrostatic potential (upper panel). ACF7-GAR or its mutant was isolated from *E. coli* as his tagged recombinant proteins. Microtubule binding was examined by co-sedimentation assay. Pellet and supernatant were immunoblotted with α-His6 to determine binding affinity. (**g**) *In vitro* FRAP analysis of WT, *ACF7* KO cells and KO cells rescued with WT or mutant *ACF7*. WT *ACF7* but not mutant *ACF7* restored tight junction dynamics ($P < 0.05$, Student's $t$-test). Sample size $n > 9$ (three independent tests, and three technical replicates each). (**h**) Scratch-wound-healing model. Quantification of the kinetics of wound closure shows WT *ACF7* but not *ACF7* mutant restores wound healing of *ACF7* KO cells *in vitro*. Error bar represents s.d. Sample size $n = 3$ (three independent tests).

**Table 1 | Data collection and refinement statistics of ACF7-GAR.**

| | Native | Hg derivative |
|---|---|---|
| *Data collection* | | |
| Space group | P212121 | P212121 |
| Cell dimensions | | |
| a, b, c (Å) | 74.04, 45.99, 25.62 | 73.28, 46.14, 25.73 |
| α, β, γ (°) | 90, 90, 90 | 90, 90, 90 |
| Resolution (Å) | 50.00–1.50 | 50.00–1.60 |
| $R_{merge}$(%) | 5.6 (11.4) | 6.0 (11.8) |
| $I/\sigma I$ | 22.4 (3.5) | 24.6 (4.0) |
| Completeness (%) | 98.9 (97.9) | 98.7 (97.2) |
| Redundancy | 4.6 (4.4) | 4.8 (4.5) |
| | | |
| *Refinement* | | |
| Resolution (Å) | 25.00–1.50 | |
| No. reflections | 15,951 | |
| $R_{work}/R_{free}$ | 18.65/20.55 | |
| No. of atoms | | |
| Protein | 657 | |
| Ligand/ion | 1 | |
| Water | 119 | |
| B-factors | | |
| Protein | 20 | |
| Ligand/ion | 12 | |
| R.m.s. deviations | | |
| Bond lengths (Å) | 0.007 | |
| Bond angles (°) | 1.136 | |

resolved its X-ray structure. The resulting experimental electron density maps revealed that the asymmetric unit of the ACF7-GAR crystal consists of one protomer. The crystal structure of ACF7-GAR was determined to a resolution of 1.5 Å (Table 1 and Supplementary Fig. 1A), revealing that the ACF7-NT molecule is folded into an elongated polypeptide that is ~32.7 Å long and ~17.9 Å wide. There is one α-helix at the amino terminus (α1), followed by five antiparallel β-strands (β1, β2, β3, β4 and β5), folding into a central β-sheet from which a flexible β-hairpin structure with a positively charged surface extrudes, and another C-terminal α-helix (α2). These features are interconnected with flexible coil loops (Fig. 2d).

The microtubules have highly negatively charged surface[35]. Thus, it is most likely that the GAR domain binds microtubules through the flexible, positively charged central β-sheet (Fig. 2e). To test our hypothesis, we mutated five positively charged amino acids at β2, β3, β4, K53, R63, R66, R69 and R75 to aromatic residues K53Y, R63Y, R66Y, R69Y and R75Y (Fig. 2e,f). Tyrosine was used to replace positively charged residues because both lysine and arginine have the aliphatic part in their side chain and could contribute to the hydrophobic interaction with other side chains within the protein. Our structural modelling reveals that such substitution should not cause an unfavourable steric crash. Consistently, both WT and mutant proteins have the same elution profile in the size exclusion chromatography (Supplementary Fig. 1B). To examine its affinity with microtubules, we carried out the microtubule co-sedimentation assay. Our results indicate that the mutation in the five residues almost completely inhibits microtubule binding with ACF7's GAR domain (Fig. 2f and Supplementary Fig. 1C).

Loss of *ACF7* impairs dynamics of tight junctions. To explore the role of ACF7 binding with microtubules in this process, we generated stable cell lines from ACF7-null Caco-2 cells to re-express ACF7 or ACF7 mutant with PiggyBac transposon system[36]. After selection, the cell lines can express ACF7 or its

mutant stably. Exogenous ACF7 and its mutant were expressed at a comparable level with the correct size.

Re-expression of WT *ACF7* in Caco-2 KO cells showed a nearly complete rescue of tight junction dynamics, as determined by FRAP analysis, and restored cell motility in the scratch-wound model (Fig. 2g,h). The results confirm the specificity of CRISPR-mediated KO of *ACF7* and the role of ACF7 in the regulation of tight junction dynamics and cell migration. By contrast, expression of *ACF7* mutant failed to rescue the dynamic feature of tight junctions *in vitro* or cell motility (Fig. 2g,h), strongly suggesting that ACF7 regulates these processes through its interaction with microtubule network.

**ACF7 deficiency altered the intestinal physiology**. To investigate the potential role of ACF7 in the gut, we generated an intestinal conditional cKO model of *ACF7* using the Cre/loxP system[29]. Western blotting confirmed the deletion of *ACF7* in the small intestine and colon of ACF7 cKO mice (Fig. 3a, Supplementary Fig. 2A–C). *ACF7* cKO animals were born at a Mendelian ratio and could survive to adulthood, but displayed a variety of abnormalities, including a smaller size, frequent loose stools and diarrhoea, and susceptibility to rectocele (Fig. 3b,c). Haematoxylin and eosin (H/E) staining showed that the size of the intestinal villi and the depth of the crypts were significantly increased in the small intestine of the *ACF7* cKO mice (Fig. 3d), while the intestinal epithelial cells in cKO animals were less well organized and polarized compared with the WT cells (Fig. 3d). The colon architecture and the epithelial morphogenesis in *ACF7* cKO mice showed similar alterations (Fig. 3d). Quantitative morphometry showed significantly increased linear crypt depth and cell number per crypt in the *ACF7* cKO mice compared with WT littermates (Fig. 3e).

To examine the potential ultrastructural changes in the intestine of *ACF7* cKO mice, we carried out a scanning electron microscopy (SEM) analysis. The SEM of the small intestine showed that there were substantially fewer villi per unit area of the small intestinal epithelium in the *ACF7* cKO mice compared with the WT mice. Moreover, the villi of the *ACF7* cKO mice exhibited a hypertrophic and irregular form compared with those of WT mice (Fig. 3f). The SEM analysis of the colon showed that there were substantially fewer valvulae conniventes and deep grooves in the colonic epithelium of the *ACF7* cKO mice compared with WT littermates (Fig. 3f). Together, our results indicate that loss of *ACF7* results in substantial changes in morphogenesis of intestine and colon epithelium.

**ACF7 cKO does not affect the terminal differentiation**. Differentiated enterocytes or colonocytes comprise the most epithelial cells in mammalian gut. The intestinal *Alkaline Phosphatase* (*ALP*) gene is only expressed in differentiated enterocytes of the small intestine. To determine the potential effects of an intestinal *ACF7* deficiency on the terminal differentiation of enterocytes, the small intestine was stained for ALP. Compared with the control, the small intestine of the *ACF7* cKO mice showed no significant alterations in ALP staining (Supplementary Fig. 3A). In the colon, the potential consequence of *ACF7* deletion on the terminal differentiation of colonocytes was determined by staining for carbonic anhydrase-1 (CA1). Loss of *ACF7* did not lead to a significant difference in the colon staining of CA1, either (Supplementary Fig. 3B). The differentiation of the intestinal epithelial cells into goblet cells was examined by staining for acidic and neutral mucins using Alcian blue and periodic-acid Schiff, respectively. Our results indicate a largely normal pattern of goblet cells in both the intestine and colon of *ACF7* cKO mice

(Supplementary Fig. 3C,D). In addition, the effects of ACF7 deletion on Paneth cell differentiation were investigated. Histological analysis also indicates the presence of Paneth cells in both WT and *ACF7* cKO intestine (Supplementary Fig. 3E). Together, these results show that ACF7 is not an essential gene for the terminal differentiation of intestinal epithelial cell lineages.

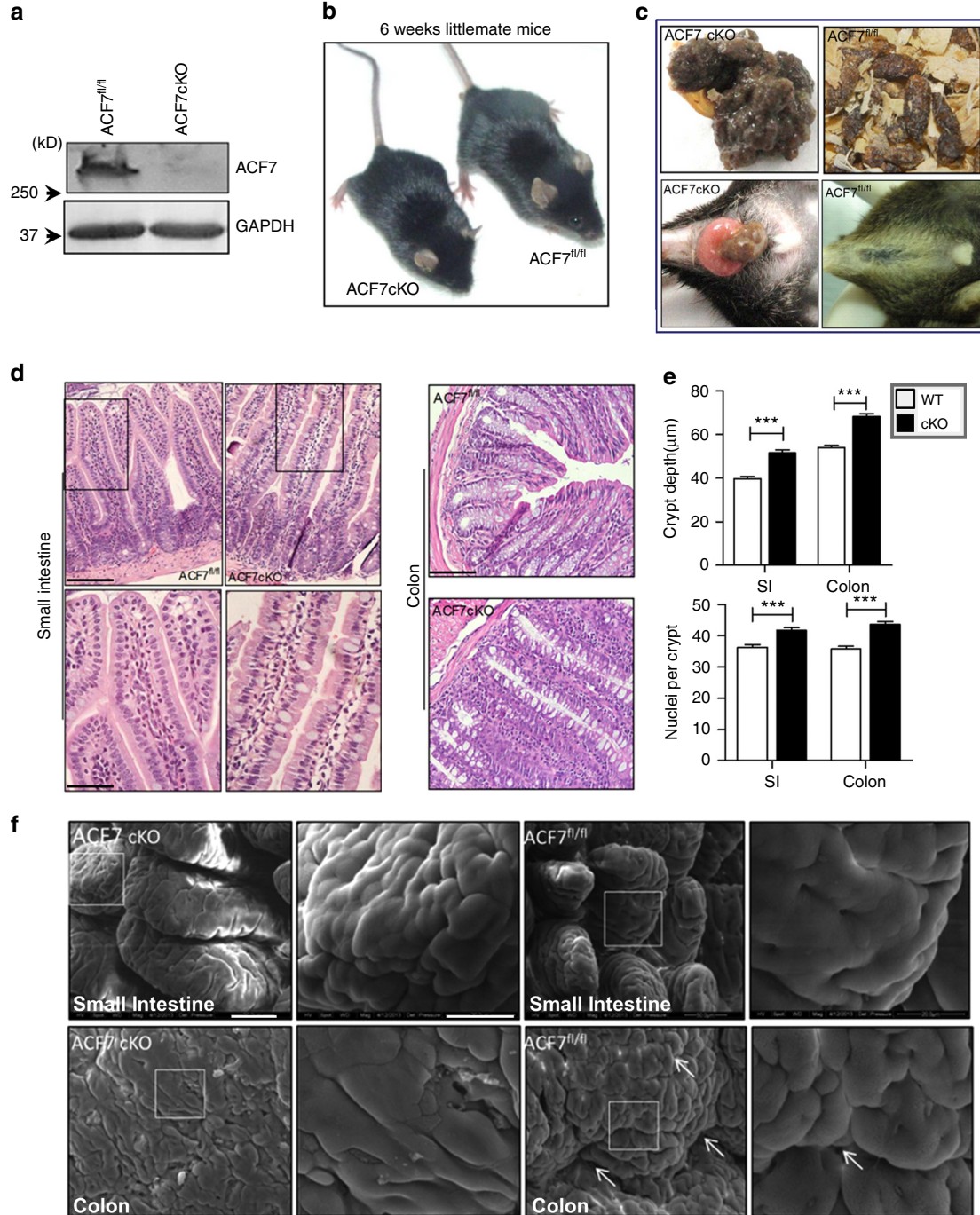

**Figure 3 | cKO of *ACF7* leads to defects in intestinal morphogenesis.** (**a**) Western blot analysis verified ACF7 deficiency in the intestine of the *ACF7* cKO mice. (**b**) Adult *ACF7* cKO mice are smaller in size. (**c**) *ACF7* cKO animals have more frequent loose stools (upper panels) and are more susceptible to rectocele (lower panels) compared with their WT littermates. (**d**) HE staining in the small intestine and colon reveals that the intestinal epithelial morphogenesis in the *ACF7* cKO mice is significantly altered compared with that in the WT mice. Boxed areas in intestine sections are enlarged and shown at the bottom. Scale bar, 200 μm (upper panels) or 50 μm (lower panels). (**e**) Linear crypt depth measurements in micrometres and cell number per crypt are significantly increased in the *ACF7* cKO mice compared with the WT mice (mean ± s.d., n = 10, ***P < 0.001, Student's t-test). SI, small intestine. Sample size n > 3 (three independent tests on different animals). (**f**) SEM of the small intestine (left panels) demonstrated that there are fewer villi per unit area in the small intestinal epithelium, and the villi are more hypertrophied and irregular in the *ACF7* cKO mice compared with the WT mice. SEM of the colon (right panels) shows that there are fewer valvulae conniventes and regularly deep grooves in the colonic epithelium of the *ACF7* cKO mice compared with the WT mice (indicated by the white arrow). Boxed areas are enlarged and shown at the right. Scale bar, 50 or 20 μm (enlarged panels).

**ACF7 regulates tight junction dynamics and wound repair.**
Accumulating evidence suggests that ACF7 regulates the cytoskeletal and cell adhesion dynamics and promotes the directional movement of cells[16,17,33,37]. Thus, we examined the effects of *ACF7* deficiency on the migration of intestinal epithelial cells *in vivo*. At 3 weeks of age, mice were pulse-labelled in the S-phase with bromodeoxyuridine (BrdU) and killed 4 or 24 h later for immunohistochemical examination. The migration rate of the epithelial cells, as defined by the height of the BrdU-positive cells along the crypt–villus axis, was significantly diminished in the intestines of the *ACF7* cKO mice compared to those of the WT littermates (Fig. 4a and quantification in Fig. 4b). Our results suggest that ACF7 regulates the movement of intestinal epithelial cells along the crypt–villus axis during normal intestinal homeostasis.

Healing of intestinal wounds is critically involved in pathogenesis of various gut diseases. To examine the potential role of ACF7 in this process, we bred our *ACF7* cKO strain with transgenic mice carrying an *mRFP-ZO1* expression cassette driven by the villin promoter[34]. Intravital imaging permitted us to monitor the wound healing of intestinal epithelium upon laser ablation damage *in vivo*. Consistent with our *in vitro* observations, deletion of *ACF7* caused a significant delay in repairing the oligocellular wound *in vivo* (Fig. 4c and quantification in Fig. 4d).

Epithelial wound healing requires significant cell shape change and extensive reorganization of cytoskeletal network and cell junctions. Despite being an enormous structure, the tight junction is dynamically regulated during physiological and pathological processes[38,39]. It has been demonstrated that tight junction components have to be dynamically translocated to the purse string structure during the healing of epithelial wounds[40]. To assess tight junction dynamics upon loss of *ACF7*, we employed a FRAP analysis *in vivo* with the *mRFP-ZO1* transgenic animals. Ablation of *ACF7* in intestinal epithelial cells led to a significant decrease in tight junction dynamics *in vivo* (Supplementary movies 3 and 4, and montage in Fig. 4e). Quantification indicated that $T_{1/2}$ for fluorescence recovery was significantly increased in *ACF7* cKO animals (Fig. 4f). Taken together, our results strongly suggest that ACF7 regulates tight junction dynamics, epithelial cell migration and wound healing in intestinal epithelium *in vivo*.

**ACF7 cKO mice are susceptible to DSS-induced colitis.**
Aberrant cellular junctions and impaired wound-healing capability could make significant contributions to pathogenesis in the digestive system, such as IBD[1,23,24,41,42]. The spontaneous diarrhoea of *ACF7* cKO animals (Fig. 3c) suggests that deletion of *ACF7* could increase the susceptibility to infection or injury of the digestive tract. To determine this possibility, we examined whether these mice were more susceptible to mucosal inflammation during colitis. *ACF7* cKO mice and their littermates were subjected to drinking water containing 2% DSS (dextran sulf sodium). Strikingly, 6 days post DSS treatment, all of the cKO mice died, whereas more than 80% of WT littermates survived the same treatment (Fig. 5a). Starting from day 2 of DSS treatment, significant body weight loss and increased disease activity index (DAI) were observed in the DSS-treated *ACF7* cKO mice compared with WT controls (Fig. 5b,c). In addition, *ACF7* cKO animals exhibited more severe faecal blood scores as well as diarrhoea score upon DSS treatment (Fig. 5d,e). At the end of the treatment (by day 7), the colon of *ACF7* cKO mice were significantly lighter and shorter compared with WT controls (Fig. 5f, and quantification in Fig. 5g). Tissue histology indicated that the entire colon and caecum of cKO animals were filled with loose and bloody stool when moribund (Fig. 5h). Taken together, our results provide compelling evidence that deletion of *ACF7*

leads to significantly increased susceptibility to DSS-induced colitis.

An aberrant inflammatory response plays a critical role in development of IBD[43]. To address the potential role of ACF7 in this process, we first analysed blood cell profiles upon DSS treatment. Haematological parameters including red and white blood cell count, haematocrit and haemoglobin in the blood of *ACF7* cKO and WT mice were measured after 4 days of DSS treatment. The *ACF7* cKO mice showed a significant decrease in red blood cells, haematocrit and haemoglobin during the acute colitis course when compared with WT littermates (Fig. 6a,b). Meanwhile, we observed a significant increase in the white blood cell count in ACF7 cKO mice compared with WT mice (Fig. 6a). No significant differences in red and white blood cell count, haematocrit and haemoglobin were observed between ACF7[fl/fl] and ACF7 cKO mice before DSS treatment (Fig. 6a,b).

DSS treatment also led to significant tissue architecture change in the colon of *ACF7* cKO mice, as determined by H&E staining (Fig. 6c). Although there was no appreciable difference in the basal inflammation scores between WT and *ACF7* cKO mice, the *ACF7* cKO animals exhibited a significant increase in histologic inflammation after DSS treatment (Fig. 6c and quantification in Fig. 6d). Additionally, ACF7 cKO colons exhibited substantially greater inflammatory cell infiltration, submucosal swelling and epithelial damage compared with ACF7[fl/fl] colons upon DSS treatment (Fig. 6e).

During colitis, there is an infiltration of immune cells accompanied by changes in cytokine gene expression, which occurs in response to the ensuing damage. CD68[+] and CD3[+] T lymphocytes are critical effectors of the mucosal immune activation. The α-CD68 immunofluorescence staining demonstrated a significant increase in numbers of monocytes in the colon of *ACF7* cKO mice compared with WT mice (Fig. 6f and Supplementary Fig. 4A). Similarly, a significant increase in CD3 positive T cells was observed in DSS-treated *ACF7* cKO mice compared with ACF7[fl/fl] mice (Fig. 6f and Supplementary Fig. 4A), suggesting sustained immune activation.

To further explore the changes in immune activation, the levels of inflammatory cytokine and chemokine, macrophage inflammatory protein-2 (MIP-2) and tumour necrosis factor (TNF)-α, were measured in a colonic organ culture system. Before DSS treatment, we did not detect any significant difference in the basal level of the inflammatory mediators (Fig. 6g). However, after DSS-induced colitis, the levels of MIP-2 ($0.219 \pm 0.020$ versus $0.133 \pm 0.011$ ng ml$^{-1}$, 95% confidence interval, 3 replicates) and tumour necrosis factor-α (TNF-α) ($0.163 \pm 0.011$ versus $0.118 \pm 0.009$ ng ml$^{-1}$) were dramatically increased in the colon of *ACF7* cKO mice compared with WT organ (Fig. 6g). Meanwhile, the expression levels of MIP-2 and TNF-α were also measured in the serum. On day 4 of DSS treatment, significant increase in the serum levels of MIP-2 ($0.053 \pm 0.015$ versus $0.013 \pm 0.002$ ng ml$^{-1}$) and TNF-α ($0.1073 \pm 0.0033$ versus $0.0956 \pm 0.0014$ ng ml$^{-1}$) were observed in the ACF7 cKO mice compared with ACF7[fl/fl] mice (Supplementary Fig. 4B). Together, these results show that *ACF7* cKO mice are highly susceptible to intestinal inflammation induced by DSS, accompanied by extensive epithelial erosion, inflammatory cell infiltration and sustained immune activation.

**Suppression of *ACF7* in human patients with UC.** To examine the role of ACF7 in the pathogenesis of UC in humans, we started by screening the potential transcriptional alteration of cytoskeletal regulators in human UC patients. With the human cytoskeleton regulator array, we identified 15 potential candidates that exhibit significant changes (more than twofold change) in

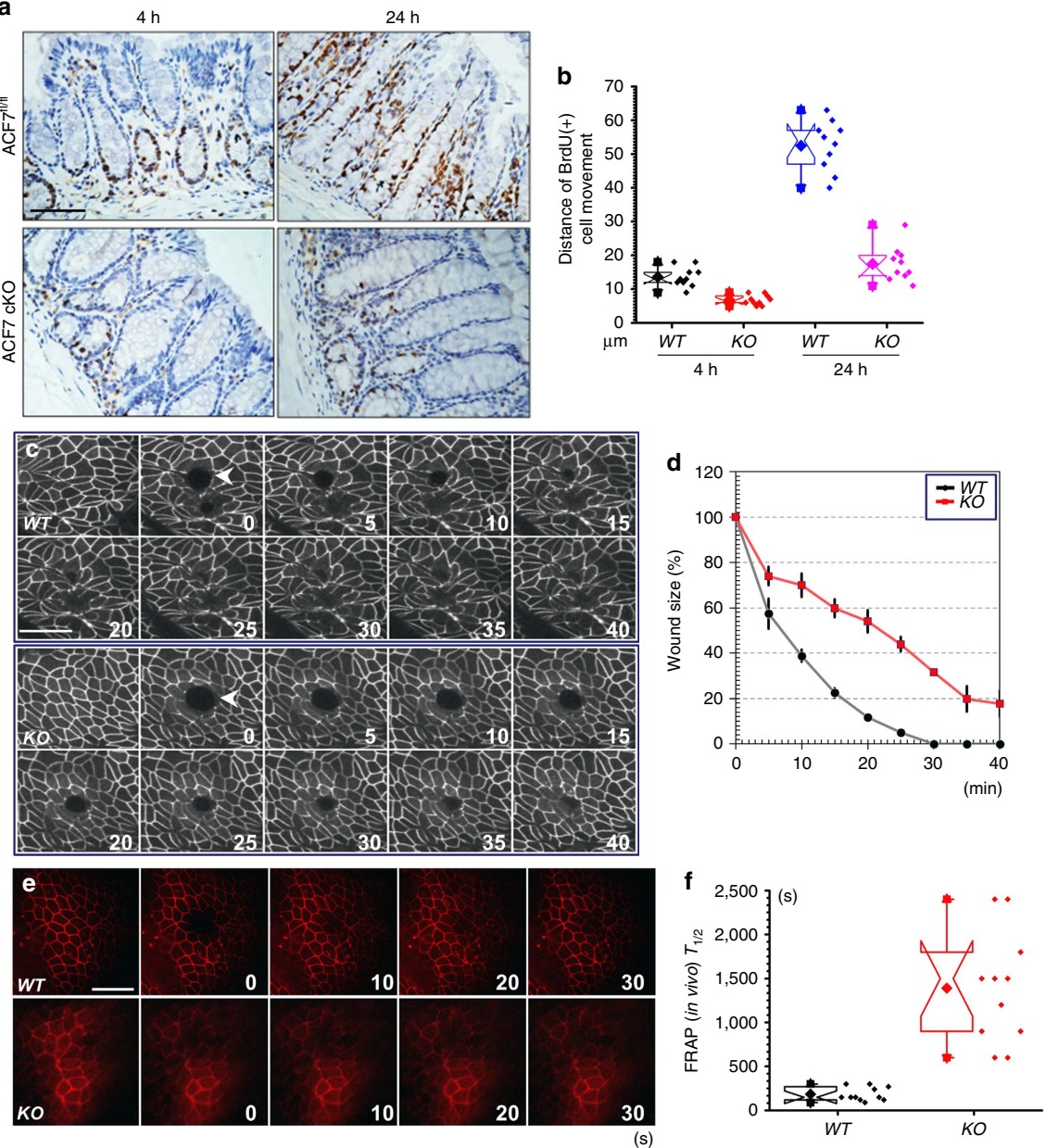

**Figure 4 | ACF7 regulates wound repair and tight junction dynamics of intestinal epithelial cells *in vivo*.** (**a**) The representative sections of the intestines from the WT and *ACF7* cKO mice at 4 and 24 h post BrdU pulse show that the migration of BrdU-positive cells from the crypt is inhibited by deletion of *ACF7*. Scale bar, 50 μm. (**b**) Quantification of BrdU-positive cell migration *in vivo*, indicating significantly decreased migration in the *ACF7* cKO mice compared with the WT mice (n = 10, ***P < 0.001, Student's *t*-test). Sample size n > 9 (three independent tests with different animals, and three technical replicates each). (**c**) Oligocellular wounds were introduced to colonic epithelium by laser ablation in *vil-mRFP-ZO1* transgenic mice with *ACF7* cKO or WT background. Would healing *in vivo* is monitored by intravital imaging with multiphoton microscopy. Scale bar, 50 μm. (**d**) Wound healing *in vivo* was quantified by measuring the percentage of wound closure over time with ImageJ. Note WT heals significantly faster than cKO mice. Sample size n = 3 (three independent tests on different animals), mean ± s.d. (**e**) FRAP analysis was carried out to determine the dynamics of tight junctions. Montages show fluorescence recovery of mRFP-ZO1 *in vivo*. Scale bar, 50 μm. (**f**) Box and whisker plotting of $T_{1/2}$ (half time) for FRAP analysis in WT and cKO colon epithelium. Deletion of *ACF7* significantly increases $T_{1/2}$ *in vivo* (P < 0.01, Student's *t*-test). Sample size n > 9 (three independent tests, and three technical replicates each).

UC patients in comparison to the control group (Supplementary Fig. 5A,B). Six genes that have more than twofold changes have been shown to play specific role in orchestrating dynamics of microtubule network, and two of them show a statistically significant difference (P < 0.05), including *CLIP2* and *ACF7* (Fig. 7a). The diminished expression of *ACF7* was confirmed by immunohistochemistry with colon biopsy samples collected from

UC patients (Fig. 7b,c). To further evaluate the role of ACF7 in UC, we collected more than 40 clinical samples from UC patients with low or relatively high expression levels of *ACF7* (Supplementary Table 1). When subjected to pathological analysis, we found that the low level of *ACF7* is significantly associated with higher histological inflammation activity, higher serum CRP (C reactive protein) level and higher rate of faecal

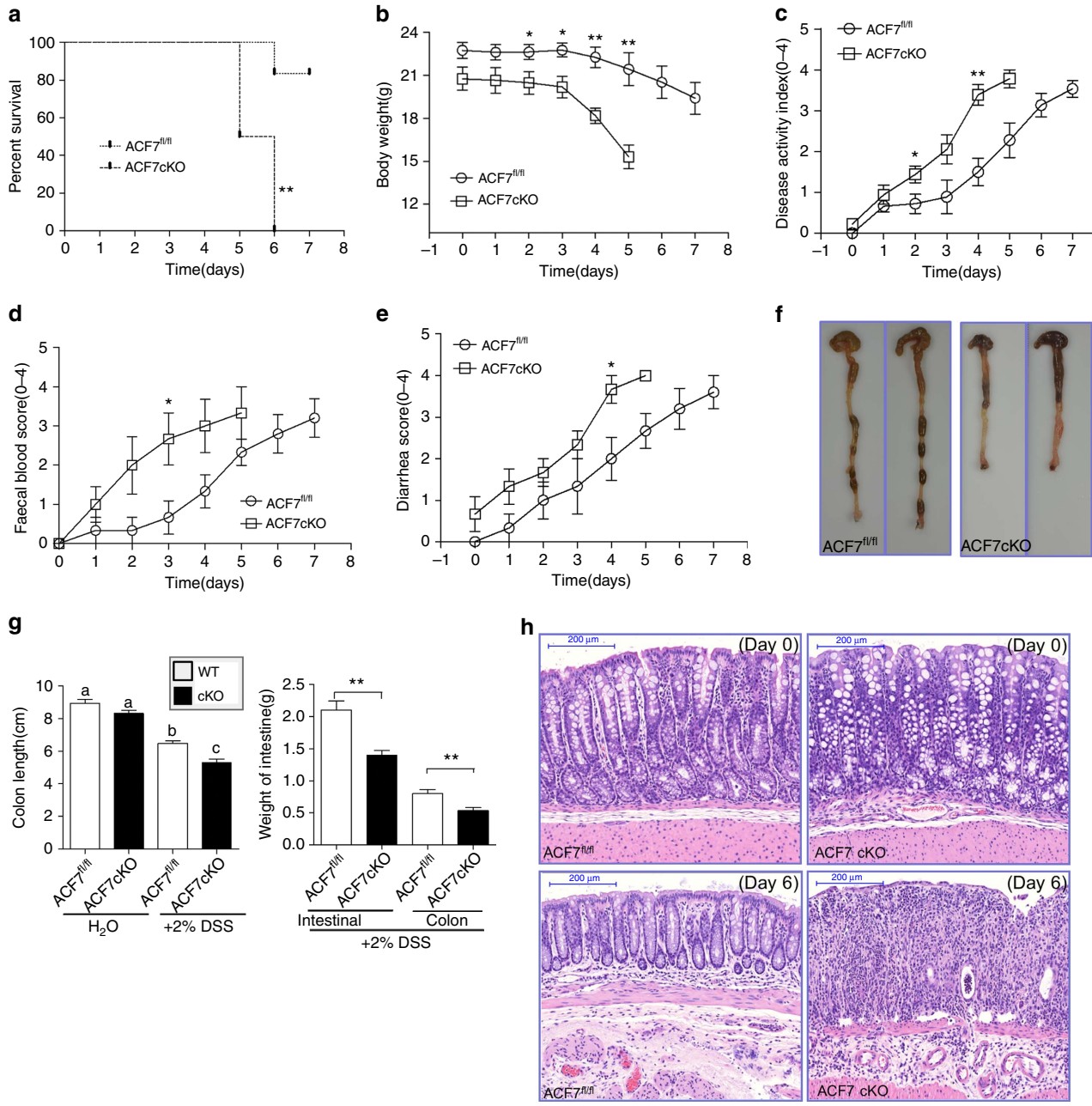

**Figure 5 | *ACF7* cKO mice are more susceptible to DSS-induced colitis.** (**a**) Kaplan–Meier survival curve of DSS treatment in *ACF7* cKO mice and WT littermates ($n = 6$ animals, **$P < 0.01$, log-rank test). (**b–e**) To quantitatively evaluate DSS-induced colitis, we examined body weight loss (**b**), disease activity index (**c**), faecal blood (**d**) and diarrhoea (**e**) in *ACF7* cKO animals and WT littermates (mean ± s.e.m., $n = 6$ per time point per group, *$P < 0.05$, **$P < 0.01$; Student's *t*-test). Note significantly enhanced disease development in cKO group. (**f,g**) The colon weight and length of WT and *ACF7* cKO mice were measured and quantified (means ± s.e.m., $n = 6$ animals, data with different superscript letters $P < 0.05$, one-way ANOVA). (**h**) Representative H/E staining images of WT and *ACF7* cKO colon at the end of DSS treatment. Scale bar, 200 μm.

occult blood in the human patients (Fig. 7d–f and Supplementary Table 1). Together, our results strongly suggest that ACF7 plays a significant role in the pathogenesis of UC in human patients.

## Discussion

Mammalian spectraplakin, such as ACF7, can crosslink microtubule and F-actin networks. In skin keratinocytes, it has been shown that ACF7 can guide microtubule plus ends towards focal adhesions and is critically involved in the regulation of focal adhesion disassembly and cell motility[7,16,17]. In this report, we demonstrated that conditional deletion of *ACF7* in intestinal epithelium resulted in impaired dynamics of tight junctions and decreased rate of intestinal epithelial cell migration and wound repair, which in turn led to increased susceptibility to experimental colitis *in vivo*. Consistent with these results, the expression level of *ACF7* significantly correlates with development of UC in human patients, suggesting that cytoskeletal coordination mediated by ACF7 could contribute to the pathogenesis of IBD by controlling cell/cell junction dynamics, intestinal epithelial cell movement and wound healing *in vivo*.

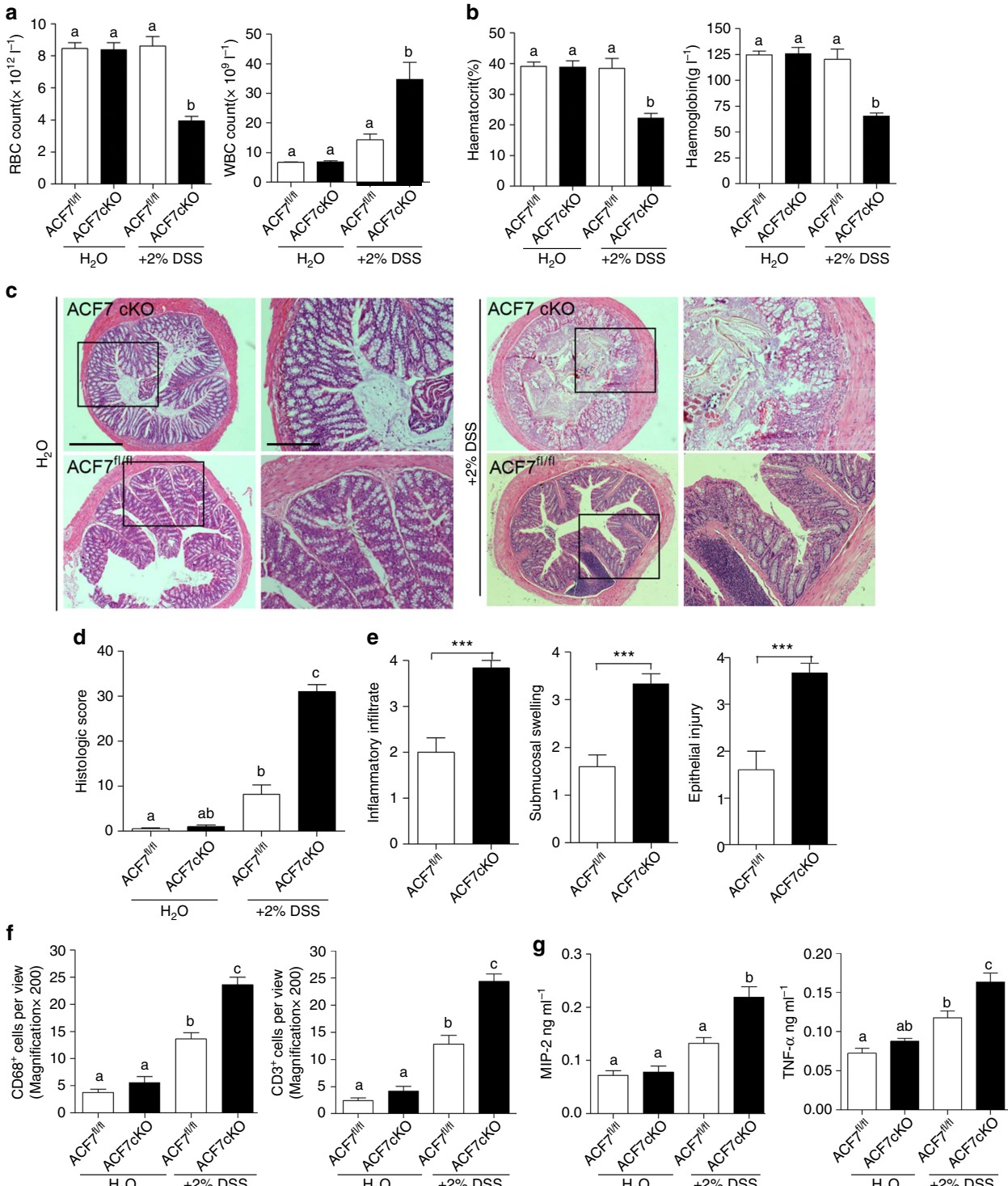

**Figure 6 | Elevated inflammatory response in *ACF7*-deficient mice. (a)** Red blood cell count (RBC) and white blood cell (WBC) counts in the blood of WT and *ACF7* cKO mice. Cell counts were measured at day 4 of DSS administration. (means ± s.e.m., $n = 6$ animals, data with different superscript letters $P < 0.05$, one-way ANOVA). **(b)** Haematocrit, haemoglobin in the blood of WT and *ACF7* cKO mice, measured at day 4 of DSS administration. (means ± s.e.m., $n = 6$ animals, data with different superscript letters $P < 0.05$, one-way ANOVA). **(c)** H&E staining showing epithelial damage and inflammation of colon in WT and *ACF7* cKO mice. Boxed areas are enlarged and shown at the right. Scale bar, 1,000 μm or 200 μm (enlarged panels). **(d)** Histologic inflammation scores were calculated for WT and *ACF7* cKO mice (means ± s.e.m., $n = 6$ animals, data with different superscript letters $P < 0.05$, one-way ANOVA). **(e)** Inflammatory infiltrate, submucosal swelling and epithelial injury were measured in DSS-treated *ACF7* cKO mice and control littermates (mean ± s.e.m., $n = 6$ animals, $***P < 0.001$, Student's $t$-test). **(f)** CD68 and CD3 immunofluorescence staining in the colon of DSS-treated *ACF7* cKO mice and WT littermates, (means ± s.e.m., $n = 6$ animals, data with different superscript letters $P < 0.05$, one-way ANOVA). **(g)** The expression levels of MIP-2 and TNF-α in colon culture supernatants in *ACF7* cKO mice and WT controls with or without DSS treatment (means ± s.e.m., $n = 6$ animals, data with different superscript letters $P < 0.05$, one-way ANOVA).

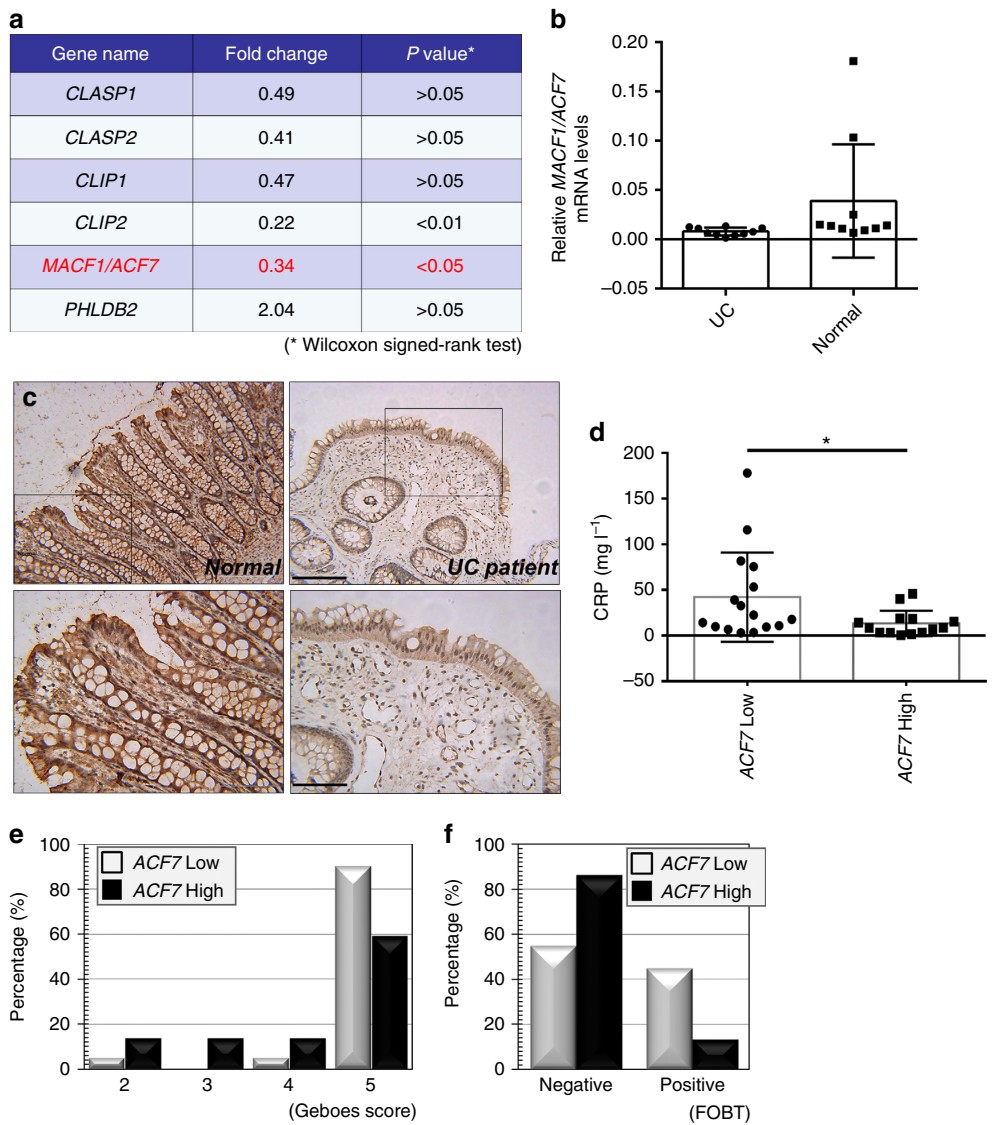

**Figure 7 | ACF7 expression *in vivo* is associated with development of UC in human patients.** (**a**) Human cytoskeleton regulator PCR array has been used to assess the transcriptional alteration in colonic epithelium of patients with ulcerative colitis (UC). The table indicates the fold changes (UC patients versus. control) and *P* value (Wilcoxon signed-rank test) of six microtubule-related genes. Note ACF7 has ~threefold decrease in UC group with $P < 0.05$. (**b**,**c**) Quantification and representative immunohistochemistry staining of ACF7 in colonic biopsy from UC patients and control (normal) group ($n = 10$ patient samples, $P < 0.05$, Student's *t*-test). Boxed area is enlarged and shown at the bottom of panel **c**. Note reduced ACF7 level in UC group. Scale bar, 200 or 50 μm (enlarged panels at bottom). (**d**–**f**) Suppressed expression of *ACF7* was correlated with enhanced presence of CRP (C reactive protein) (**d**) increased Geboes score (**e**) and increased FOBT (faecal occult-blood test) (**f**) in UC patients. ($n = 10$ patient samples, $P < 0.05$, Student's *t*-test). Data in panels **b** and **d** are mean ± s.d.

The tight junction is a major component of intestinal barrier function, creating a boundary between the apical and basolateral membranes and acting as a primary barrier to the diffusion of solutes through the intercellular space[23,24]. However, instead of acting as a simple barrier for intestinal epithelium, tight junctions are highly dynamic structures whose assembly, disassembly and degree of sealing vary according to various external stimuli and physiological and pathological conditions[38]. The dynamic regulation of tight junctions is essential for embryonic morphogenesis, tissue remodelling and epithelial to mesenchymal transition. Although not completely understood, accumulating evidence suggest that the remodelling of the epithelial apical junctional complex, including both adherence junction and tight junction, is regulated by two major mechanisms involving the internalization of junctional proteins and the

reorganization of peri-junctional F-actin network[2,38]. Several important signalling molecules have been implicated in tight junction dynamics, including protein kinases such as protein kinase C (PKC), and small GTPases, such as Rho[38,44,45].

Tight junctions associate with the F-actin network in epithelial cells. Accumulating evidence also demonstrate the interaction of microtubules with tight junction. The association of tight junctions with non-centrosomal microtubules in the planar apical networks has been shown to be mediated by the AMPK (monophosphate-activated protein kinase) phosphorylation of cingulin, which plays a critical role in epithelial morphogenesis *in vitro*[3]. The interaction of microtubules with cell/cell adhering junctions can serve as a spatial cue for microtubule polarity and the dynamic arrangement of microtubule networks in cells. On the other hand, it has also been shown that microtubules

regulate the integrity of adherence junctions and tight junctions. Disruption of the microtubule network by drugs like colchicine can inhibit the epithelial barrier function in thyroid epithelial and lung endothelial cells[46,47]. Additionally, treatment of various epithelial cells with nocodazole has been shown to impair the dynamics of adherence junctions and tight junctions and prevent their disassembly in a classical extracellular calcium depletion model[6], reminiscent of the effect of microtubule disruption on focal adhesion stabilization[16,31,32]. In this study, we demonstrated that ACF7 can promote microtubule association with tight junctions in intestinal epithelial cells, and disruption of the microtubule network or deletion of *ACF7* inhibits tight junction dynamics and impairs intestinal wound repair, suggesting that the dynamics of cell/cell junctions and focal adhesions may be regulated via similar mechanisms, involving spectraplakin-mediated cytoskeletal coordination.

It remains elusive how guided microtubule dynamics may participate in cell/cell junction dynamics. It has been shown that microtubules may regulate adherence junction through the polarized delivery of cadherins. With *Drosophila* embryo as a model, studies have indicated that microtubules can enhance the assembly of adhesions by sustaining Par3 and E-cadherin level at adherence junctions through a dynein-dependent apical restriction[48]. Microtubules can also elevate the dynamic Par3/E-cadherin pool by inhibition of Rho GTPase[49]. It is more unclear how microtubule dynamics may contribute to the disassembly of cell junctions. Endocytosis has been implicated in the turnover of both focal adhesions and cell/cell junctions[2,50]. Our recent study demonstrated a new signalling cascade, suggesting that microtubules can deliver protein kinase MAP4K4 (mitogen-activated protein kinase kinase kinase kinase 4) to focal adhesions, which in turn phosphorylates IQSEC1 (IQ motif and SEC7 domain-containing protein 1) and activates the Arf6-mediated endocytic pathway to enhance focal adhesion disassembly[51]. It will be interesting to determine whether a similar pathway functions to mediate microtubule-dependent tight junction dynamics in the future.

Abnormalities in cell junctions and aberrant cell movement and wound healing are critically involved in the pathogenesis of IBD[1,23,24,41,42]. In this study, we demonstrated an essential role of ACF7 in the regulation of tight junction dynamics, epithelial cell movement and wound healing in intestinal epithelium both *in vitro* and *in vivo*. With cKO of *ACF7*, we demonstrated that loss of *ACF7* can exacerbate experimental colitis in mice. Consistently, reduced expression level of *ACF7* correlates with the development of UC in human patients. Together, this reveals an intriguing connection between spectraplakin-controlled cytoskeletal/cell adhesion dynamics and the development of IBD. However, the exact mechanisms whereby cytoskeletal and cell adhesion coordination contributes to colitis remain unclear at present. Interestingly, aside from ACF7's well-established role in cytoskeletal orchestration, ACF7 can also participate in cellular signal transduction[7]. For instance, it has been shown that ACF7 regulates Wnt signalling, and loss of *ACF7* affects the expression of various Wnt downstream targets[10]. Thus, to fully understand its role in intestinal pathogenesis, it will also be important to examine the potential changes in the transcriptome of *ACF7* cKO epithelium. Future studies will be essential to delineate the potential signalling cascade involved in this process.

In closing, our findings provide critical insights into the mechanics behind ACF7-dependent cytoskeletal coordination in tight junction dynamics and intestinal cell movement, paving the way for probing more deeply into the intricate signalling network orchestrating the crosstalk between F-actin and microtubule cytoskeletal networks in intestinal epithelium, as well as its relevance in the physiology and pathology of human gut.

## Methods

**Reagents and plasmid DNA constructions.** Rabbit serum against BrdU was produced by BD Pharmingen (560810, dilution 1:200). Rabbit serum against pan-ACF7 (dilution 1:2,500) was produced by Covance company (Princeton, NJ). Rabbit polyclonal Abs against HA (sc-805, dilution 1:1,000) and His tag (sc-803, dilution 1:1,000) were obtained from Santa Cruz Biotechnology, Inc. (Santa Cruz, CA). BrdU was produced by Sigma. DSS (molecular weight 36,000–50,000 Da) was obtained from MP Biomedicals company. Fluorescein isothiocyanate-dextran (FITC-Dextran) (average mol wt 4,000 kDa) was obtained from Sigma-Aldrich (USA). Rabbit polyclonal anti-CD3 was obtained from Epitomics (3256-1, dilution 1:100) and anti-CD68 (rabbit polyclonal) was produced by Abcam company (ab125212, dilution 1:100). Enzyme-linked immunosorbent assay kits of TNF-α and MIP-2 were produced by EIAab Science Co., Ltd (Wuhan,China). EDTA-containing tubes were produced by Axygen, Inc., Union City (California, USA). Complete blood count determined in an Sysmex xs-800i blood analysis system (Sysmex Corporation, Japan). Analysis for FITC-Dextran concentration with a Synergy HT Multi-Mode Microplate Reader (BioTek, USA) at an excitation wavelength of 485 nm and emission wavelength of 535 nm.

**Animals.** Mice homozygous for the germline floxed (flanking loxP, mixed background) insertions were bred to Villin-Cre (Vil-Cre) recombinase transgenic mice (Jackson Laboratories, USA)[52], which efficiently excised floxed exons. Neonatal mice genotypic for Vil-Cre and ACF7$^{fl/fl}$ alleles (cKO) were born in the expected Mendelian numbers. The cKO mice were further crossed with vil-RFP-ZO1 mice for intravital imaging of intestinal epithelium and tight junctions. The mice were housed under specific pathogen-free conditions in the University of Chicago and Affiliated Sixth People's Hospital of Shanghai Jiaotong University and consumed a standard sterile diet and filtered water *ad libitum* under a 12-h light–dark cycle. The experimental protocol was approved by the Animal Care and Use Committee and the Ethics Committee of University of Chicago and Shanghai Jiaotong University.

**Induction of DSS-induced colitis.** DSS (MP Biomedicals) was dissolved in the drinking water of mice. Fresh DSS solution was provided every day. 8- to 12-week-old male ACF7$^{fl/fl}$ ($n = 6$) and ACF7 cKO ($n = 6$) mice were exposed to 2.0% DSS for 7 days. Meanwhile, 10- to 14-week-old female ACF7$^{fl/fl}$ ($n = 6$) and ACF7 cKO ($n = 6$) mice were exposed to 2.0% DSS for 5 days followed by normal drinking water for 5 days. The mice were checked each day for morbidity, and weight was recorded. Induction of colitis was determined by weight loss, faecal blood, diarrhoea, and, upon autopsy, weight and length of colon. To quantify induction of colitis, a DAI was calculated as reported[53–57]. DAI was calculated for each mouse daily based on body weight loss, occult blood and stool consistency/diarrhoea. A score of 1–4 was given for each parameter, with a maximum DAI score of 12. Score 0: no weight loss, normal stool, no blood; score 1: 1–3% weight loss; score 2: 3–6% weight loss, loose stool, blood visible in stool; score 3:6–9% weight loss; score 4: >9% weight loss, diarrhoea, gross breeding. Loose stool was defined as the formation of a stool that readily becomes paste upon handling. Diarrhoea was defined as no stool formation. Gross bleeding was defined as fresh blood on fur around the anus with extensive blood in the stool.

**Intestinal permeability *in vivo*.** This measure is based on the intestinal permeability towards fluorescein isothiocyanate-dextran (FITC-Dextran) (Sigma-Aldrich, USA) as described[58–60]. Briefly, mice that fasted for 6 h were given FITC-Dextran by gavage (0.6 mg g$^{-1}$ body weight, 50 mg ml$^{-1}$). After 1 h, 120 μl of blood was collected from the tip of the tail vein in mice before DSS treatment. After DSS administrated, we collected 120 μl blood by cardiac puncture after given FITC-Dextran 1 h when end of the experiment or time of dying. The blood was centrifuged at 4 °C, 10,000 g for 5 min. Plasma was diluted in an equal volume of PBS (pH 7.4) and analysed for FITC-Dextran concentration with a Synergy HT Multi-Mode Microplate Reader (BioTek; USA) at an excitation wavelength of 485 nm and emission wavelength of 535 nm. Standard curves were obtained by diluting FITC-dextran in non-treated plasma diluted with PBS (1:3 v/v).

**Histopathological analysis of mouse colon tissue.** For each animal, histological examination was performed on the colon; samples were fixed in 10% formalin before staining with H&E. All histological quantitation was performed blinded using a scoring system as described[56,57,61]. The three independent parameters measured were severity of inflammation (0–3: none, slight, moderate, severe), extent of injury (0–3: none, mucosal, mucosal and submucosal, transmural) and crypt damage (0–4:none, basal one-third damaged, basal two-thirds damaged, only surface epithelium intact, entire crypt and epithelium lost). The score of each parameter was multiplied by a factor reflecting the percentage of tissue involvement ( × 1, 0%–25%; × 2, 26%–50%; × 3, 51%–75%; × 4, 76%–100%) and all numbers were summed. The maximum possible score was 40.

**Intravital imaging of mice and FRAP.** Intestinal wound-healing and FRAP analysis were carried out in live animals[34,62,63]. For FRAP analysis, mice were

developed and imaged with an SP5 microscope (Leica) with a $40 \times$ NA 0.8 oil immersion objective. mRFP-ZO1 was excited and bleached using the DPSS 561 laser. For wound-healing analysis, mice were anaesthetized and the abdomens were opened by a midline incision. Electrocautery was used to open a 2-CM loop of jejunum along the antimesenteric border. The abdominal cavity was closed under the loop of jejunum while protecting the neurovascular supply. The mucosal surface of the jejunum was placed against the coverslip bottom of a petridish containing HBSS. Mice were imaged with multiphoton confocal microscope in the light imaging facility of University of Chicago.

**Immunofluorescence analyses.** Formalin-fixed, paraffin- embedded colon sections were deparaffinized in xylene and rehydrated with alcohol. Samples were boiled for 10 min in antigen retrieval solution (SLNco, CinoAsia, China) and left at room temperature for 30 min. Slides were then incubated (1 h, room temperature) with primary antibodies for CD3 (1:100, Epitomics, Callifornia, USA), CD68 (1:100, Abcam, UK). After three times washing with PBS, the slides were incubated for 45 min, room temperature with FITC labelled secondary antibodies (Jackson Immuno, Darmstadt, Germany). Slides were microscopic examined and photographed (Nikon Eclipse 80i, Japan).

**Colon organ culture and enzyme-linked immunosorbent assay.** Colon tissue (about 1 cm) from rectum to caecum were excised, opened and cut longitudinally washed in cold phosphate-buffered saline (PBS) supplemented with penicillin, streptomycin and amphotericin B and incubated in 1.5 ml serum-free RPMI 1,640 medium, penicillin, streptomycin and amphotericin B cultured in 12-well, flat-bottom culture plates (Corning Incorporated, NY, USA) at $37\,^{\circ}C$ in $5\%\ CO_2$. After 24 h, supernatant fluid was collected, centrifuged and stored at $-80\,^{\circ}C$. Supernatants were analysed for TNF-$\alpha$, macrophage inflammatory protein-2 (MIP-2) content in duplicate using commercially available enzyme-linked immunosorbent assay kits (EIAab Science Co., Ltd, Wuhan, China). In some experiments, serum was obtained by centrifugation, stored at $-80\,^{\circ}C$, and then cytokine concentration was measured.

**Haematological analysis.** Hundred microlitres of blood was collected from the tip of the tail vein on fourth day of 2% DSS treatment, blood was drawn into EDTA-containing tubes (Axygen, INC. Union City, California, USA). After vigorous mixing, complete blood count was determined in an Sysmex xs-800i blood analysis system (Sysmex Corporation, Japan) using mouse specific settings. White blood cell (WBC) population was determined by blood smear, and corrected by WBC count.

**Patients' sample collection.** The project was approved by the institutional review board of the Shanghai Tenth People's Hospital (affiliated with Tongji University, Shanghai, China). Written informed consent was obtained from all the patients and tissue sample donors, and anonymity was assured by tracking the patients through their clinical history numbers.

**qRT-PCR and western analysis.** Total RNA was isolated with the RNA Extraction Kit (SLNco, Cinoasia, China). The first strand synthesis of cDNA was performed with the Reverse Transcription Kit (TOYOBO, Japan). Reactions were run on the FTC-3000 Qrt-PCR machine (Funglyn, Canada). Relative changes in mRNA and expression were determined using the $2-\Delta\Delta Ct$ method[64]. Q-PCR and qPCR-array assays were carried out by xybiotech (http://www.xybiotech.com) using the Human Cytoskeleton Regulator PCR array platform with the company's proprietary primer set.

Proteins were extracted by scraping the luminal side of the intestine onto a clean glass slide using another clean glass slide. The scraped tissue was then homogenized in lysis buffer containing 50 mM tris-HCl (pH 6.8) and 2% sodium dodecyl sulfate (SDS). Insoluble material was removed by centrifugation at $20,000g$, and the supernatant was collected for protein quantification and SDS gel electrophoresis. Following protein transfer, the membrane was immunoblotted with different antibodies. The signals were detected bychemiluminescence.

**Immunohistochemical analyses.** Dissected tissues were fixed in 4% paraf-ormaldehyde solution and paraffin-embedded (FFPE). Five-micrometre sections were dewaxed in xylene and rehydrated in graded alcohol baths. The slides were treated with methanol with $0.3\%\ H_2O_2$ to block endogenous peroxidase activity, then microwave antigen retrieval was performed for 20 min in citrate buffer (pH 6.0) determined. After washing in 10 mM Tris buffer (pH 6.0), the slides were incubated with 3% normal sheep serum to eliminate non-specific staining and then incubated with primary antibody for 30 min at $37\,^{\circ}C$. For immunostaining, the peroxidase-conjugated secondary antibody reagent was used and the peroxidase activities of staining were visualized with 3,3'-diaminobenzidine-tetra-hydrochloride (DAB, Sigma-Aldrich, St. Louis, MO, USA). The nuclei were counter-stained with haematoxylin. MACF1 antibodies were purchased from Sigma. The immunostaining grading was performed based on the staining extent (percentage of positive cells graded on scale from 0 to 3:0, none; 1,1–30%; 2, 31–60%; 3, >60%) and the staining intensity (graded on scale from 0 to 3: 0,

none; 1, mild/weak; 2, moderate; 3, strong). The combination of extent ($E$) and intensity ($I$) of staining ($E \times I$) was gained varying from 0 to 9.

The slides were stained with H/E for histological examination before immunostaining. To evaluate the disease severity of UC patients, the Geboes grading system was used[65].

**Cell culture and CRISPR-mediated KO.** Caco-2 cells were obtained from Dr. Jerrold Turner lab and maintained in DME (4.5 g glucose per liter) with 10% FBS. Cells were cultured at $37\,^{\circ}C$ incubator with $5\%\ CO_2$. For CRISPR KO, cells were infected with lentivirus encoding both Cas9 and gRNA specifically designed for human ACF7. Potential contamination with mycoplasma was screened using the ATCC universal mycoplasma detection kit.

**Confocal microscopy.** WT or ACF7 KO caco-2 cells were fixed and stained. Images were taken on Leica SP8 confocal system. 3D reconstruction and surface 2D plot were carried out with NIH Image J software.

**Cloning, expression and purification of ACF7-GAR.** The ACF7-GAR containing a $6 \times$ His tag at the N-terminus was overproduced in *Escherichia coli* BL21(DE3) cells. Briefly, cells were grown in 10 l of LB medium containing 100 μg ml$^{-1}$ ampicillin at $30\,^{\circ}C$. The culture was induced with 0.4 mM isopropyl-β, D-thiogalactopyranoside (IPTG) at an $OD_{600}$ (optical density at 600 nm) value of $\sim 0.5$. Cells were collected within 5 h of induction. The collected bacteria were resuspended in ice-cold buffer containing 20 mM Na–Hepes (pH 7.5), 100 mM NaCl and 1 mM PMSF. The cells were then lysed in a French pressure cell. Cell debris was removed by centrifugation for 1 h at $4\,^{\circ}C$ and $25,000g$. The protein solution was then purified with $Ni^{2+}$-affinity and G-200 sizing columns. The purity of the protein ($>95\%$) was judged using 10% SDS-PAGE stained with Coomassie Brilliant Blue. The N-terminal sequence of the ACF7-GAR protein was confirmed by sequencing. The purified ACF7-GAR protein was then concentrated to 20 mg ml$^{-1}$ in a buffer containing 25 mM Hepes pH 7.5, 150 mM NaCl, 2 mM $CaCl_2$ for crystallization.

**Crystallization of GAR domain.** Crystals of the $6 \times$ His ACF7-GAR were obtained using hanging-drop vapour diffusion. The GAR crystals were grown at $4\,^{\circ}C$ in 24-well plates with the following procedures. One-microlitre protein solution containing 20 mg ml$^{-1}$ ACF7-GAR protein was mixed with 1 μl of reservoir solution containing 0.1 M sodium cacodylate trihydrate pH 6.5, 0.2 M $(NH_4)_2SO_4$, 25%(v/v)PEG4000. The resultant mixture was equilibrated against 500 μl of the reservoir solution. Typically, the dimensions of the crystals were $0.2\,mm \times 0.1\,mm \times 0.1\,mm$. All crystals were flash frozen with 25% glycerol in mother liquor for checking and data collection. Crystals were derivatized in mother liquor solution by soaking with 0.1 M Hg(AC)$_2$ for 2 h before flash frozen, back soaking was also applied to improve the clarity.

**Data collection and structure determination.** GAR crystals were tested and screened on an in-house X-ray source of Rigaku MicroMax-002 + system equipped with Saturn 944 + CCD detector (Rigaku) and Oxford Cryo-system. The data sets were integrated and processed using Rigaku crystalclear-1.44 for in-house data sets and HKL2000 for Synchrotron. High-resolution native data sets were collected at 100 K on beamline BL17U of Shanghai Synchrotron Radiation Facility, the crystal structure of GAR was determined by single isomorphous replacement with anomalous scattering of mercury derivative and refined to 1.5 Å.

Heavy atoms searching and initial Phasing were undertaken by *SHELXC/D/E* and atomic model was auto-built *PHENIX* suite, model improvement and refinements were finalized in *COOT* and *REFMAC* in *CCP4*. In the final model, the electron density is clear and continuous, 98.5% residues located in favoured region, 2.5% located in allowed region and no residues located in outlier region by Ramachandran plot. All crystallographic figures were drawn in Pymol.

**Biochemical analysis.** Microtubule binding was examined using microtubule binding protein spin-down assay kit from Cytoskeleton (Denver, CO), according to manufacturer's instructions. In brief, about 0.1–0.9 nM of assembled microtubules were incubated with about GAR wt/mutant molecules ($\sim 1\,\mu g$) in a 200 μl reaction. The bound and unbound proteins were separated by ultracentrifugation. Western blotting analysis was performed as described previously[66]. Uncropped versions of western blots are in Supplementary Fig. 6.

**Statistical analysis.** Statistical analyses were performed using the GraphPad Prism software (GraphPad Prism Software, Version 5.01, GraphPad, San Diego, CA) and SPSS for Windows 15.0.0 (SPSS, Inc., USA) statistical software. Survival curves were estimated using the Kaplan–Meier method, and the resulting curves were compared using the log-rank test. For comparisons, a one-way analysis of variance and Student's t-tests (two-tailed) were performed as appropriate. Correlation analyses were performed using Pearson's test. Statistical significance was defined as $P < 0.05$.

**Data availability.** For this study, we will make our data available to the scientific community, which will avoid unintentional duplication of research. All the research data will be shared openly and timely in accordance with the most recent NIH guidelines. The structure results have been deposited to Protein Data Bank (PDB) with accession code 5 × 57.

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

## Acknowledgements

We thank Dr Elaine Fuchs at the Rockefeller University for sharing reagents. We thank Dr. Vytas Bindokas at the University of Chicago Light Microscopy Core, Linda Degenstein at Univesity of Chicago Transgenic Core Facility and Dr. Lara Leoni at the University of Chicago Animal Imaging facilities for their excellent technical assistance. The animal studies were carried out in both the ALAAC-accredited animal research facility at the University of Chicago and animal facility at the Shanghai Jiaotong University. This work was supported by a grant R01-AR063630 from the National Institutes of Health, the Research Scholar Grant (RSG-13-198-01) from the American Cancer Society, the Janet D. Rowley Discovery Fund award, and the V scholar award from V foundation to X.W. This work was supported by grants AA021434 and AA020265 from the National Institutes of Health to S.C. This work was also supported by the Natural Science Foundation of China (31460232; 1600410; 810010698), Natural Science Foundation of Guangxi (2013GXNSFGA019010, 2014GXNSFDA118016) and High-level innovation team and distinguished scholar programme of Guangxi universities to F.Y., Innovation Project of Guanxi Graduate Education 2017 (Design metal drugs targeting tumour cell migration related proteins) to Y.Z. and F.Y.

## Author contributions

Y.M., J.R.T., W.-J.T., F.Y., H.L., H.Q. and X.W. designed the experiments. Y.M., J.Y., Y.Z., C.S., M.O., W.G.L., Q.W., A.G. and X.G. performed the experiments. Y.M., J.Y., Y.Z., C.S., M.O., W.G.L., Q.W., S.-Y.C., J.Z. and W.-J.T. analysed the data. X.W. wrote the manuscript. All authors edited the manuscript.

## Additional information

**Competing interests:** The authors declare no competing financial interests.

DOI: 10.1038/ncomms16121    OPEN

# Erratum: ACF7 regulates inflammatory colitis and intestinal wound response by orchestrating tight junction dynamics

Yanlei Ma, Jiping Yue, Yao Zhang, Chenzhang Shi, Matt Odenwald, Wenguang G. Liang, Qing Wei, Ajay Goel, Xuewen Gou, Jamie Zhang, Shao-Yu Chen, Wei-Jen Tang, Jerrold R. Turner, Feng Yang, Hong Liang, Huanlong Qin & Xiaoyang Wu

*Nature Communications* 8:15375 doi: 10.1038/ncomms15375 (2017); Published 25 May 2017; Updated 11 Jul 2017

The affiliation details for Yanlei Ma and Yao Zhang are incorrect in this Article. The correct affiliation details for these authors are given below:

Yanlei Ma:
Department of GI surgery, Shanghai Tenth People's Hospital Affiliated with Tongji University, 301 Yanchang Road, Shanghai 200072, China.
The University of Chicago, Ben May Department for Cancer Research, Chicago, Illinois 60637, USA.
Department of Colorectal Surgery, Fudan University Shanghai Cancer Center, Shanghai, China.

Yao Zhang:
State Key Laboratory Cultivation Base for the Chemistry and Molecular Engineering of Medicinal Resources, Ministry of Science and Technology of China, Guanxi Normal University, Guilin 541004, China.

