## [Peer Review File · Nature Communications]

Reviewers' comments:

Reviewer #1, and expert in the epithelial barrier and colitis (Remarks to the Author):

The paper by Ma et al. reports that ACF7 stabilizes tight junctions and may be important in diseases like ulcerative colitis.

The paper requires minor revisions before publications.

The paper (especially the results) should be written in past tense.

The title should be modified to: ACF7 a crosslinker of microtubule and F-actin regulates intestinal ...by orchestrating tight junction dynamics.

The address should be consistent:
TX instead of Texas; KY instead of Kentucky
Medical School (capital M and S)

State alternate names of ACF7 in the Introduction: MACF1, macrophin, etc.

There are minor grammatical errors in the text. Proof read once more.

Percentage of DSS used should be stated in the results section.
Figure 7F: Bar graph spell check: Positive

The paper is novel and will be of interest to researchers in gastroenterology. The paper may lead to development of drugs that target the tight junctions, alteration of which is known to induce UC.

The paper is recommended for publication in Nature Communications.

Reviewer #2, and expert in crystallography (Remarks to the Author):

This manuscript by Ma et al., using a combination of X-ray crystallography, biochemistry and cell biology characterises the role of ACF7, a microtubule binding protein, in regulating intestinal response to epithelial injury. They conclude that ACF7 functions by modulating microtubule and tight junction dynamics.

Some of the key components of this manuscript are i) the structural characterisation of ACF7 GAR domain and its interaction with microtubules and ii) evaluation of the role of ACF7-MT interactions on tight junction dynamics. The structure of MT-binding GAR domain of ACF7 has been determined at high resolution. Using electrostatic surface potential analysis they identified putative MT-binding interface. Mutating positively charged residues of this putative interface affects MT-binding in vitro and tight junction dynamics in vivo. Overall the quality of the data presented is good. But I do have some concerns, which need to be addressed to make this MS suitable for publication.

1. It is not clear how the surface charge potential was calculated? There are different ways to calculate electrostatics. The popular model is implemented in APBS. The authors need to clearly mention how electrostatics were calculated? what were the assumptions?

2. To probe the role of this interface, they have mutated positively charged residues to Tyr residues. It is not clear what is the rationale for mutating positively charged residues to Tyr, a bulky residue that also has hydrophobic property. Generally, positively charged residues are

mutated to negatively charged residues such as Asp/Glu. The authors need to clarify this and show that mutations do not affect the overall folding of the domain - by comparing the size exclusion profiles of wt and mutant GAR domain or by performing Circular Dichroism.

3. Authors did not describe the experimental details of MT-cosedimentation assay - What is the concentration of protein and MT used? if possible they should perform MT-cosedimentation assay with varying concentration of MTs and measure binding affinity of WT and mutant protein?

4. From the structure determination table, it looks like there may still some usable high resolution data. R_{sym} and I/SigI for the high resolution bin is about 11 and 3.5, which would mean that crystals diffract beyond 1.5Å. I wonder why they have restricted their analysis to 1.5Å.

5. I also have a general comment - At many places authors instead of citing original articles tend to cite review articles. It would be great if they pay more attention to how they cite previous studies.

Reviewer #3, and expert in tight junctions (Remarks to the Author):

I report on the study "ACF7 regulates intestinal response to epithelial injury by orchestrating microtubule and tight junction dynamics" from the Xiaoyang Wu lab. This and other labs have published on ACF7 in high-ranked journals before, one in Nat. Commun. from the same group already this year (not cited).

The present study is technically very well done and I have no issues on this score. The statistical methods are appropriate.

My first homework was to understand what's new compared to other studies on ACF7, where effects on microtubules, tight junction proteins, epithelial permeability, morphogenesis, cell migration, and wound healing were reported before. In fact it opens, besides other aspects, one exciting new perspective of ACF7, namely its concerted action on microtubule regulation and tight junction dynamics in relation to wound healing. Therefore it is no surprise that only this aspect is reflected in the Title of the manuscript.

New is also the increased inflammatory response in ACF7-deficient mice. Finally, in patients suffering from ulcerative colitis (UC) ACF7 expression was found elevated.

My only major issue on this ms. is that it sets a big sweeping blow on all aspects of ACF7 and thus is not detailed enough regarding each single topic. For example, the functional and mechanistic implications of ACF7 and ulcerative colitis remain unclear. Which one is the hen and which one the egg? The simple correlation between ACF7 and UC has been shown before (Schwan et al. 2009). This important topic should perhaps be cancelled here and investigated in more detail separately.

Other points:

Does ACF7 deficiency affect transcription of regulators of the cytoskeleton? This cannot be proven in patients, because UC is caused by numerous other factors besides ACF7.

The paper of Yue et al. 2016, Nat. Commun. DOI: 10.1038/ncomms11692 "In vivo epidermal migration requires focal adhesion targeting of ACF7" is not referenced, but should be set in context with the present study.

Importantly, the paper of Liang et al. 2013 Int. J. Mol. Med. "ACF7 regulates colonic permeability" is not considered. It showed for the first time an effect of ACF7 knockout on the tight junction proteins claudin-1 and occludin in the colon.

Overall Comments:

We were delighted that the reviewers found our work novel and interesting to the readership of *Nature communications*. Each reviewer had enormously helpful comments. We've now fully addressed these issues, and in doing so, have substantially improved the paper and its impact. We've conducted the various experiments suggested by each reviewer and revised the manuscript accordingly as we delineate below (all major changes are highlighted in the left margin). We really thank these reviewers for all of their constructive comments!

Reviewers' comments:

Reviewer #1, and expert in the epithelial barrier and colitis (Remarks to the Author):

The paper by Ma et al. reports that ACF7 stabilizes tight junctions and may be important in diseases like ulcerative colitis.

The paper requires minor revisions before publications.

We thank the reviewer for appreciating the novelty and importance of our work.

The paper (especially the results) should be written in past tense.

We thank the reviewer for pointing it out. The manuscript has been revised accordingly, and has been proofread by a native speaker.

The title should be modified to: ACF7 a crosslinker of microtubule and F-actin regulates intestinal ...by orchestrating tight junction dynamics.

We changed the title as suggested.

The address should be consistent:
TX instead of Texas; KY instead of Kentucky
Medical School (capital M and S)

We changed the affiliations as suggested.

State alternate names of ACF7 in the Introduction: MACF1, macrophin, etc.

Included as suggested (page 2, first paragraph).

There are minor grammatical errors in the text. Proof read once more.

The manuscript has been proofread by a native English speaker.

Percentage of DSS used should be stated in the results section.

Included as suggested

Figure 7F: Bar graph spell check: Positive

Corrected.

The paper is novel and will be of interest to researchers in gastroenterology. The paper may lead to development of drugs that target the tight junctions, alteration of which is known to induce UC.

The paper is recommended for publication in Nature Communications.

Reviewer #2, and expert in crystallography (Remarks to the Author):

This manuscript by Ma et al., using a combination of X-ray crystallography, biochemistry and cell biology characterises the role of ACF7, a microtubule binding protein, in regulating intestinal response to epithelial injury. They conclude that ACF7 functions by modulating microtubule and tight junction dynamics.

Some of the key components of this manuscript are i) the structural characterisation of ACF7 GAR domain and its interaction with microtubules and ii) evaluation of the role of ACF7-MT interactions on tight junction dynamics. The structure of MT-binding GAR domain of ACF7 has been determined at high resolution. Using electrostatic surface potential analysis they identified putative MT-binding interface. Mutating positively charged residues of this putative interface affects MT-binding in vitro and tight junction dynamics in vivo. Overall the quality of the data presented is good. But I do have some concerns, which need to be addressed to make this MS suitable for publication.

We thank the reviewer for the constructive comments on our manuscript.

1. It is not clear how the surface charge potential was calculated? There are different ways to calculate electrostatics. The popular model is implemented in APBS. The authors need to clearly mention how electrostatics were calculated? what were the assumptions?

The surface charge potential was calculated by APBS (Baker et al., 2001) after prepared by PDB2PQR server (Dolinsky et al., 2004) and such information is added to the legend of Figure 2.

2. To probe the role of this interface, they have mutated positively charged residues to Tyr residues. It is not clear what is the rationale for mutating positively charged residues to Tyr, a bulky residue that also has hydrophobic property. Generally, positively charged residues are mutated to negatively charged residues such as Asp/Glu. The authors need to clarify this and show that mutations do not affect the overall folding of the domain - by comparing the size exclusion profiles of wt and mutant GAR domain or by performing Circular Dichroism.

We chose to mutate these residues to tyrosine based on our structure. We agree with this reviewer that it is logical to replace lysine and arginine to glutamate or aspartate for the charge reversal. However, both lysine and arginine also have the aliphatic part in their side chain and could contribute the hydrophobic interaction with other side chains within the protein. This is the case for arginine 66, 69, and 75. We found that arginine 66 and 75 sandwich the side chain of tryptophan 80 while arginine 66 and 69 sandwich that of leucine 68. Thus, we need to preserve such hydrophobic interactions to maintain the structure integrity. We chose tyrosine because it preserves the requisite hydrophobic interactions with leucine 68 and tryptophan 80 while its hydroxyl group allows the favorable interaction with solvent. To be consistent, we also change the other two residues to tyrosine. Our structural modeling reveals that such substitution should not cause the unfavorable steric clash. We have performed the size exclusion chromatography as suggested this reviewer. Our result showed that the mutant protein had the same elution profile as the wild type construct (Supplementary Fig. 1A).

We have also revised the Result section to include this discussion (Page 6 second paragraph).

3. Authors did not describe the experimental details of MT-cosedimentation assay - What is the concentration of protein and MT used? if possible they should perform MT-cosedimentation assay with varying concentration of MTs and measure binding affinity of WT and mutant protein?

MT (microtubule) cosedimentation assay was performed following manufacturer's protocol (Cytoskeleton Inc.). In brief, about 1×10^{11} assembled MT molecules were incubated with about GAR wt/mutant molecules (~1 μ g) in a 200 μ l reaction. The final concentrations of MT and GAR wt/mutant proteins are ~0.4nM and 0.6nM respectively. The details are included in the Material & Methods in the revised manuscript.

We have also carried out the cosedimentation assay with varying concentration of MTs as suggested (Supplementary Fig. 1B). The results are consistent with our original finding, and strongly suggest that mutations on the positively charged residues will diminish MT interaction.

4. From the structure determination table, it looks like there may still some usable high resolution data. R_{sym} and I/SigI for the high resolution bin is about 11 and 3.5, which would mean that crystals diffract beyond 1.5Å. I wonder why they have restricted their analysis to 1.5Å.

We thank the reviewer for pointing out this issue. The crystal diffracted beyond 1.5 Å, however the diffraction data of higher resolution was not collected due to restrictions on the detector distance. When we tried to scale our data to 1.4 Å, it significantly decreased the completeness of the data (0% for 1.48-1.45 Å).

5. I also have a general comment - At many places authors instead of citing original articles tend to cite review articles. It would be great if they pay more attention to how they cite previous studies.

We thank the reviewer for pointing this issue out. We have included more citations with original literature as suggested in the revised manuscript.

Reviewer #3, and expert in tight junctions (Remarks to the Author):

I report on the study "ACF7 regulates intestinal response to epithelial injury by orchestrating microtubule and tight junction dynamics" from the Xiaoyang Wu lab. This and other labs have published on ACF7 in high-ranked journals before, one in Nat. Commun. from the same group already this year (not cited).

We thank the reviewer for pointing it out. We have included this paper in our reference.

The present study is technically very well done and I have no issues on this score. The statistical methods are appropriate.

We thank the reviewer for the encouraging comments.

My first homework was to understand what's new compared to other studies on ACF7, where effects on microtubules, tight junction proteins, epithelial permeability, morphogenesis, cell migration, and wound healing were reported before. In fact it opens, besides other aspects, one exciting new perspective of ACF7, namely its concerted action on microtubule regulation and tight junction dynamics in relation to wound healing. Therefore it is no surprise that only this aspect is reflected in the Title of the manuscript.

New is also the increased inflammatory response in ACF7-deficient mice. Finally, in patients suffering from ulcerative colitis (UC) ACF7 expression was found elevated.

My only major issue on this ms. is that it sets a big sweeping blow on all aspects of ACF7 and thus is not detailed enough regarding each single topic. For example, the functional and mechanistic implications of ACF7 and ulcerative colitis remain unclear. Which one is the hen and which one the egg? The simple correlation between ACF7 and UC has been shown before (Schwan et al. 2009). This important topic should perhaps be cancelled here and investigated in more detail separately.

We are highly grateful to the reviewer for appreciating the novelty of our work and the constructive comments. We agree with the reviewer that our current work covers both the role of ACF7 on cytoskeletal/tight junction dynamics and its potential function in intestinal epithelium that regulates inflammatory and injury response of intestinal epithelial cells, and contributes to development of ulcerative colitis.

With cKO animal model of ACF7, we provided convincing evidence that loss of *ACF7* will increase experimental colitis in the mice, which also correlates with reduced *ACF7* expression in human UC patients. Additionally, we provided *in vivo* evidence that loss of *ACF7* can lead to defective migration of intestinal epithelial cells, suppressed wound healing and aberrant tight junction dynamics in intestinal epithelium. It together provides strong evidence that ACF7 contributes to development of inflammatory intestinal diseases such as colitis through regulation of cell junctional dynamics, cell movement, and wound healing. However, we agree with the reviewer, because of the scope of a single study, we cannot delve further deeper into the potential mechanisms how cytoskeletal and tight junction dynamics controlled by ACF7 contributes to the *in vivo* phenotype. It is certainly an important question, and we are very interested to address it in our future studies. The discussion section of the manuscript has been revised to address this issue (page 14, second paragraph). We have also revised the title to better represent all the findings that we made in this manuscript.

Other points:

Does ACF7 deficiency affect transcription of regulators of the cytoskeleton? This cannot be proven in patients, because UC is caused by numerous other factors besides ACF7.

We agree with the reviewer that deletion of *ACF7* could lead to potential transcriptional changes. For instance, it has been shown that ACF7 regulates Wnt signaling and loss of *ACF7* affects expression of various Wnt downstream targets (Chen et al., 2006). Thus, to fully understand its role in intestinal pathogenesis, it will also be important to examine the potential changes in the transcriptome of *ACF7* cKO epithelium.

We have revised the Discussion to include this issue (Page 14, second paragraph).

The paper of Yue et al. 2016, Nat. Commun. DOI: 10.1038/ncomms11692 "In vivo epidermal migration requires focal adhesion targeting of ACF7" is not referenced, but should be set in context with the present study.

We cited this paper as suggested.

Importantly, the paper of Liang et al. 2013 Int. J. Mol. Med. "ACF7 regulates colonic permeability" is not considered. It showed for the first time an effect of ACF7 knockout on the tight junction proteins claudin-1 and occludin in the colon.

We have cited and discussed this work in the revised manuscript (page 4, second paragraph).

Reference:

Baker, N.A., Sept, D., Joseph, S., Holst, M.J., and McCammon, J.A. (2001). Electrostatics of nanosystems: application to microtubules and the ribosome. Proc Natl Acad Sci U S A 98, 10037-10041.

Chen, H.J., Lin, C.M., Lin, C.S., Perez-Olle, R., Leung, C.L., and Liem, R.K. (2006). The role of microtubule actin cross-linking factor 1 (MACF1) in the Wnt signaling pathway. Genes Dev 20, 1933-1945.

Dolinsky, T.J., Nielsen, J.E., McCammon, J.A., and Baker, N.A. (2004). PDB2PQR: an automated pipeline for the setup of Poisson-Boltzmann electrostatics calculations. Nucleic acids research 32, W665-667.

REVIEWERS' COMMENTS:

Reviewer #2 (Remarks to the Author):

Authors have addressed all the reviewers comments adequately. I am happy to recommend this manuscript for publication in Nature Communications.